# Soft mode origin of charge ordering in superconducting kagome CsV$_3$Sb$_5$

Philippa Helen McGuinness [1,12], Fabian Henssler [1,12], Manex Alkorta [2,3], Mark Joachim Graf von Westarp [1,4], Artem Korshunov [5], Alexei Bosak [5], Daisuke Ishikawa [6,7], Alfred Q. R. Baron [6,7], Michael Merz [1,8], Amir-Abbas Haghighirad [1], Maia G. Vergniory [9,10,11], Sofia-Michaela Souliou [1], Rolf Heid [1], Ion Errea [2,3,9] & Matthieu Le Tacon [1] ✉

Charge-density-wave (CDW) order and superconductivity coexist in the kagome metals AV$_3$Sb$_5$ (A=K, Cs, Rb), raising fundamental questions about the mechanisms driving their intertwined phases. Here we combine high-resolution inelastic X-ray scattering with first-principles calculations to uncover the origin of CDW formation in CsV$_3$Sb$_5$. Guided by structure factor analysis, we identify a soft phonon mode along the reciprocal *M-L* direction, with the strongest effect at the *L* point, where the elastic scattering intensity also grows most rapidly upon cooling. First-principles calculations incorporating lattice anharmonicity and electron-phonon coupling reproduce these observations and establish a soft-mode instability at the *L* point as the driving mechanism of CDW formation. Despite the weakly first-order character of the transition, our results unambiguously demonstrate that the CDW in CsV$_3$Sb$_5$ originates from a softened phonon, clarifying its microscopic origin and highlighting the central role of lattice dynamics in kagome metals.

The two-dimensional kagome lattice, built from corner sharing triangles and long recognized as a paradigmatic frustrated geometry[1], has emerged as a fertile ground for exotic quantum phenomena[2,3]. The recently discovered kagome metals AV$_3$Sb$_5$ (A = K, Cs, Rb) have therefore sparked intense interest. Their electronic structure is remarkably rich, hosting Dirac cones[4], flat bands[5], multiple van Hove singularities[3,6] and even a non-trivial $\mathbb{Z}_2$ topological invariant[7]. On top of this unusual band topology, these compounds exhibit a charge-density-wave (CDW) transition, at $T_{CDW} \approx 94$ K for CsV$_3$Sb$_5$, followed, at lower temperature, by superconductivity with $T_c = 2.5$ K[7]. Although the coexistence of a CDW and superconductivity is not unique to kagome metals, in this family it acquires a particularly intriguing character, providing a compelling platform to explore how geometry, topology, and electronic correlations conspire to produce intertwined quantum orders.

Despite substantial effort, however, the nature and origins of the CDW in CsV$_3$Sb$_5$ remain heavily disputed. Both $2 \times 2 \times 2$[8–10] and $2 \times 2 \times 4$[11] superstructures have been reported as well as coexistence[12] and transitions between these orders as a function of temperature[13,14] and even as a function of the sample cooling rate[12]. The CDW transition temperature and wavevector, as well as the superconductivity, also exhibit a strong sensitivity to tuning parameters such as pressure[15–18] and chemical substitution[19–24].

In addition, the mechanism behind the formation of the CDW remains unknown. A CDW can be driven by a Peierls-like instability

[1]Institute for Quantum Materials and Technologies, Karlsruhe Institute of Technology, Karlsruhe, Germany. [2]Centro de Física de Materiales (CFM-MPC), CSIC-UPV/EHU, Donostia, Spain. [3]Department of Applied Physics, University of the Basque Country (UPV/EHU), Donostia, Spain. [4]Max Planck Institute for Solid State Research, Stuttgart, Germany. [5]ESRF, The European Synchrotron, Grenoble, France. [6]Materials Dynamics Laboratory, RIKEN SPring-8 Center, Sayo, Hyogo, Japan. [7]Precision Spectroscopy Division, SPring-8/JASRI, Sayo, Hyogo, Japan. [8]Karlsruhe Nano Micro Facility (KNMFi), Karlsruhe Institute of Technology, Karlsruhe, Germany. [9]Donostia International Physics Center (DIPC), Donostia, Spain. [10]Département de Physique et Institut Quantique, Université de Sherbrooke, Sherbrooke, QC, Canada. [11]Regroupement Québécois sur les Matériaux de Pointe (RQMP), Québec, Canada. [12]These authors contributed equally: Philippa Helen McGuinness, Fabian Henssler. ✉e-mail: matthieu.letacon@kit.edu

determined by Fermi surface nesting[25,26] or correlation, such as momentum-dependent electron-phonon coupling (EPC)[27,28]. In CsV$_3$Sb$_5$, assuming a nearest-neighbor tight-binding model, the CDW wavevector coincides with the nesting vector between reciprocal $M$ points, which host nearby van Hove singularities (VHSs)[6]. Therefore, some studies have suggested that the nesting of the $M$ points is responsible for the formation of the CDW[29–34]. Others, however, considering that the fermiology is not that simple, have proposed that EPC is likely responsible[35–38]. In fact, ARPES[39,40], Raman scattering[36,41,42] and infrared spectroscopy[43] studies have all found signatures of substantial EPC in these materials.

Multiple harmonic calculations of the phonon dispersion have shown that the high symmetry phase of CsV$_3$Sb$_5$ is dynamically unstable with an instability along the $M$-$L$ direction[29,41,44]. In principle, inelastic scattering should offer an ideal method to investigate whether specific phonon branches are indeed unstable at these or other wave vectors. However, several studies have found no unambiguous evidence for sizable phonon anomalies associated with the CDW formation. In particular, neither inelastic X-ray scattering (IXS) studies[9,45] nor an inelastic neutron scattering study[46] found evidence of phonon anomalies in AV$_3$Sb$_5$ above $T_{CDW}$. Recent theoretical work has shown that the stability of the high-symmetry phase can only be understood by accounting for ionic entropy and anharmonicity, suggesting that, despite the weakly first-order character of the CDW, a strong phonon renormalization occurs in the system[37] and that the substantial broadening of the soft phonon due to scattering might render it unobservable[38].

To resolve this apparent contradiction and uncover the mechanism underlying CDW formation in CsV$_3$Sb$_5$, we performed a new series of experiments guided by calculations of the dynamical structure factor, allowing us to identify Brillouin zones where the unstable phonon branch carries measurable intensity. These revealed a pronounced softening of a phonon branch along the entire $M$-$L$ direction in reciprocal space, with the strongest effect at the $L$ point, where the associated elastic intensity also grows most rapidly upon cooling[45]. First-principles calculations, incorporating ionic fluctuations and a non-perturbative treatment of anharmonicity, reproduce these observations along with the correct transition temperature[37]. Together, our results provide unambiguous evidence that the CDW in CsV$_3$Sb$_5$ is driven by a phonon instability centered at the $L$ point and emphasize the pivotal role of quantum anharmonic effects in shaping the remarkable phase diagram of these kagome materials.

## Results and discussion

We begin by examining the harmonic phonon spectrum of the high-temperature $P6/mmm$ phase of CsV$_3$Sb$_5$, calculated using density functional perturbation theory (DFPT) and based on the experimental room-temperature lattice parameters (see ref. [47] for details). Consistent with earlier studies[29,44,48], the phonon dispersion exhibits instabilities near the high-symmetry $M$ and $L$ points, reflected by imaginary frequencies plotted as negative values in Fig. 1. One branch displays a pronounced instability along the entire $M$–$L$ direction. This anomalous behavior forms the central focus of our experimental investigation. The inelastic scattering intensity associated with a given phonon mode varies strongly across different Brillouin zones (BZ). To optimize the experimental geometry and ensure sufficient signal for the mode of interest, we calculated the dynamical structure factor in all experimentally-accessible zones to identify the most favorable scattering conditions. The results of these calculations are shown in Fig. 1a, c for two representative BZ centered at $\Gamma_{420}$ and $\Gamma_{103}$, respectively, with the full phonon dispersion overlaid in white.

The point $\mathbf{Q} = (3.5, 2.5, 0.5)$ corresponds to an $L$ point within the $\Gamma_{420}$ zone, where previous IXS measurements have been performed[45]. However, the structure factor of the unstable branch at this location is vanishingly small, suggesting that the anomalous phonon mode would be essentially unobservable there. By contrast, in the $\Gamma_{103}$ zone, the

structure factor is substantial at both the $M$ point $\mathbf{Q} = (0.5, 0.5, 3)$ and the $L$ point $\mathbf{Q} = (0.5, 0.5, 2.5)$, indicating favorable conditions for detecting this mode. Notably, such favorable scattering conditions are rare: fewer than 10% of BZs exhibit an $L$ point structure factor for this branch that reaches even half the intensity calculated at $\mathbf{Q} = (0.5, 0.5, 2.5)$ (the results of structure factor calculations for the unstable branch at various $M$ and $L$-points are shown in the Supplementary Information). These conclusions are robust against the inclusion of anharmonic effects, which stabilize the otherwise unstable phonon branches above $T_{CDW}$[37,38], and the impact of the electron-phonon scattering on the phonon linewidth. This is illustrated in Fig. 1b,d, where we show the phonon spectral weight along the $M$-$L$ direction in the two Brillouin zones considered, explicitly accounting for both anharmonicity and electron-phonon interactions.

The importance of BZ selection is further confirmed by thermal diffuse scattering (TDS) measurements. Figure 2 shows parts of the reconstructed ($HK$ 0.5) and ($HK$ 2.5) planes at 95 K, i.e. just above $T_{CDW}$. As previously reported[45], diffuse streaks along the $A$–$L$ direction (such as indicated by the purple oval in Fig. 2b) emerge below 150 K, with intensity peaking at $L$ and sharpening into super-lattice peaks at the CDW transition.

The intensity of the diffuse precursors to the CDW satellites vanishes in certain BZs. For instance, it is absent in the $A$–$L$ path from $\mathbf{Q} = (4, 2, 0.5)$ to $\mathbf{Q} = (3.5, 2.5, 0.5)$ (purple oval in Fig. 2a)—in agreement with the negligible structure factor for the unstable branch at the $\Gamma_{420}$ zone (Fig. 1a) and the absence of phonon anomalies at this $L$ point[45]. Similarly, the diffuse streak is only weakly visible around the $L$ point at $\mathbf{Q} = (3.5, 0, 0.5)$ (Fig. 2c), where no phonon anomalies are observed at the corresponding $M$ point[9]. By contrast, substantial diffuse intensity is observed around the $L$ point $\mathbf{Q} = (2.5, 0.5, 0.5)$; however no phonon softening was detected at this point in previous IXS experiments conducted in a lower-resolution (3 meV) configuration[45].

Notably, the enhanced precursor signal observed around the $L$ point at $\mathbf{Q} = (0.5, 0.5, 2.5)$ (Fig. 2b), together with the favorable structure factor for the unstable branch in the corresponding $\Gamma_{103}$ zone (Fig. 1b), suggests increased inelastic spectral weight relative to other BZs, motivating our targeted inelastic measurements for definitive confirmation.

Figure 3a–c shows raw, high-resolution (1.1 meV)[49,50] IXS spectra recorded between 97 K and 302 K at the $M$ point $\mathbf{Q} = (0.5, 0.5, 3)$, at a low-symmetry point $\mathbf{Q} = (0.26, 0.87, 2.34)$, and at the $L$ point $\mathbf{Q} = (0.5, 0.5, 2.5)$ in the $\Gamma_{103}$ zone. Upon approaching $T_{CDW}$, the low-energy spectral weight increases at the $M$ point and—even more markedly—at the $L$ point. In particular, the $L$ point shows a clear enhancement of low-energy spectral weight upon cooling (Fig. 3d). A similar but weaker effect is observed at the $M$ point.

Thanks to the high energy resolution, we can confirm that this enhancement exceeds a simple increase of the resolution-limited elastic scattering intensity (Fig. 3d), consistent with the presence of a softening phonon mode. In lower-resolution measurements the inelastic contribution to this spectral-weight enhancement is less clearly discernible (see Supplementary Note 11).

In contrast, the low-symmetry point shows no anomalies beyond the expected reduction in phonon intensity with decreasing temperature due to the Bose factor. This is confirmed by comparing the Bose-corrected spectra recorded at 97 K and 302 K at the low-symmetry point (and at several other points measured at different $L$ values), shown in Fig. 3e. As expected, the Bose-corrected intensity is essentially temperature independent. Along the $M$–$L$ line, however, a pronounced redistribution of spectral weight is observed between the 302 K and 97 K Bose-corrected spectra, with a clear loss of intensity around 11 meV and a strong enhancement below 5 meV, indicating a phonon mode evolution directly visible without any fitting procedure.

In order to provide more insight as to the precise nature of these spectral changes, we fit the data shown in Fig. 3 using the measured resolution function for the elastic peak and a set of damped harmonic

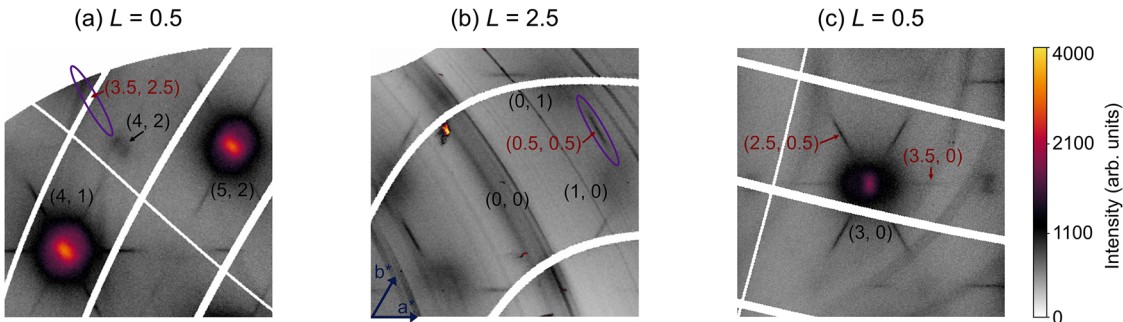

**Fig. 1 | Calculated phonon intensities in $\Gamma_{420}$ and $\Gamma_{103}$ for both harmonic and anharmonic calculations. a,c** The calculated dynamic structure factor from harmonic DFPT calculations along the line $\Gamma$-$M$-$L$ in a) $\Gamma_{420}$ c) $\Gamma_{103}$. The harmonic phonon frequencies are marked with a thin white line. **b,d** Spectral weight along the line $M$-$L$ as calculated within the SSCHA formalism at 100 K (see Supplementary Information for details), which includes both the structure factor as well as a realistic broadening of the phonon peaks due to anharmonic and electron-phonon interactions, for b) $\Gamma_{420}$ d) $\Gamma_{103}$. The position of the expected peak of the spectral function is marked with a thin white line as a reference.

**Fig. 2 | Thermal diffuse scattering just above $T_{CDW}$. a–c** The thermal diffuse scattering at 95 K in (a) $\Gamma_{420}$ (b) $\Gamma_{103}$, where all of the facets of the hexagon are crystallographically equivalent, (c) $\Gamma_{300}$. Some $A$ and $L$ points are indicated in black and red text, respectively, labeled by (H,K). The purple ovals in (a)/(b) indicate the presence/absence of diffuse streaks along the $A$-$L$ direction of the adjacent Brillouin zones.

oscillators (DHOs) for the phonons, convoluted with the resolution. The fitting approach and the functions used are described in more detail in the Methods section. The individual fitted phonons and elastic line are shown for a series of temperatures at the $M$ (Fig. 4a–d) and $L$

points (Fig. 4e–h). At the $M$ point (resp. $L$ point), there are four (resp. five) fitted phonons.

As shown in Fig. 4i, j, most fitted phonon branches are essentially temperature independent, with no discernible change in energy or

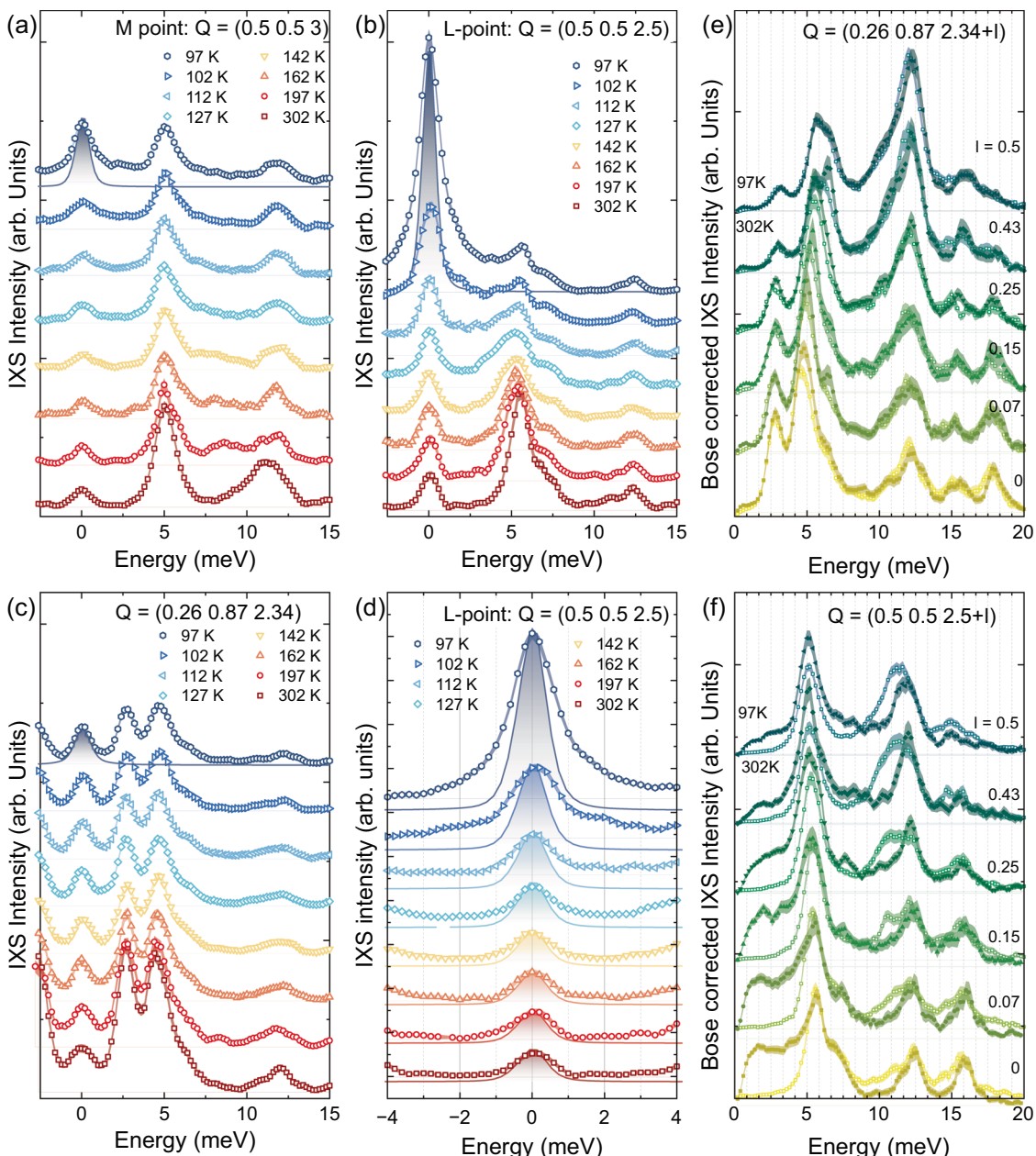

**Fig. 3 | Raw IXS spectra. a–c** Non-normalized IXS spectra, vertically offset for clarity, measured at temperatures between 97 K and 302 K at **a** the M point **Q** = (0.5, 0.5, 3), **b** the *L* point **Q** = (0.5, 0.5, 2.5), **c** a low-symmetry point **Q** = (0.26, 0.87, 2.34). The y-axis scale is not common across the subfigures. The measured instrumental resolution, scaled to the 97 K data, is shown by shaded solid lines. **d** A magnified view of the low-energy range of (**b**) with overlays of the measured instrumental resolution shaded solid lines) scaled separately to the 302 K and 97 K data, showing that the enhanced low-energy spectral weight upon cooling does not simply consist of an increase of the resolution-limited elastic peak. **e** Bose corrected IXS data at 302 and 97 K measured along the **Q** = (0.26, 0.87, 2.34 + *l*) direction (**f**) Bose corrected IXS data at 302 and 97 K measured along the *M − L* direction **Q** = (0.5, 0.5, 2.5 + *l*) direction. The thickness of the lines in all the panels represent the statistical error bars.

linewidth beyond the experimental uncertainty. A notable exception occurs at the *M* point (Fig. 4i), where a branch at 10 meV at 302 K softens dramatically to 6 meV at 97 K, just above $T_{CDW}$. The effect is even stronger at the *L* point (Fig. 4j), where the same branch softens below 3 meV at 102 K, representing an exceptionally large renormalization. At 97 K, the large elastic line prevents to reliably extract the energy of this mode, as it approaches the overdamped regime.

Such strong temperature dependence is only reproduced when anharmonic effects are explicitly included in lattice dynamics calculations. In solids, anharmonicity gives rise to thermal expansion and governs the phonon-phonon interactions responsible for the temperature-dependent frequency shifts and linewidth changes, in addition to the temperature-independent effects of EPC. These are not included in DFPT, which treats phonons within a harmonic framework at *T* = 0 K. They can, however, be incorporated non-perturbatively using the stochastic self-consistent harmonic approximation (SSCHA)[51,52], albeit at a significantly higher computational cost. Using this approach, we find that most phonons remain temperature independent (see Supplementary Information), with the notable exceptions of a $M_1^+$ and two $L_2^-$ modes, shown in Fig. 4i, j. The $M_1^+$ and one of the two $L_2^-$ modes belong to the branch which is unstable in the harmonic calculation but is stabilized by anharmonicity (see also Fig. 1c, d) and which displays strong softening upon cooling[38], consistent with the experiment. The theory also predicts an avoided crossing (see

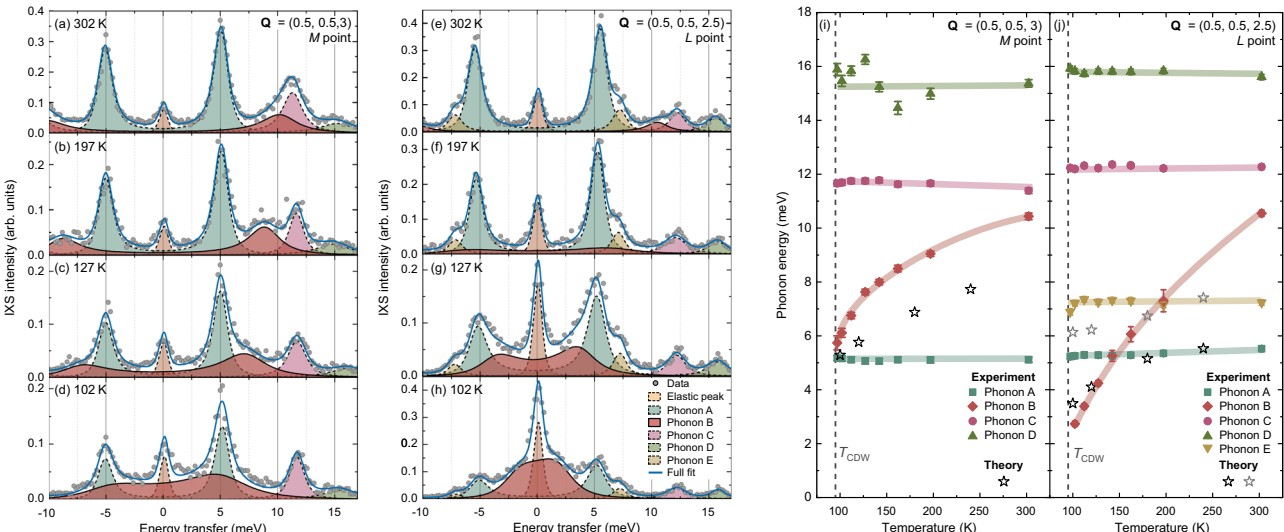

**Fig. 4 | Fitted IXS spectra and phonon energies at the *M* and *L* points as a function of temperature.** The y-axis scale is not common among the panels. **a–h** Fitted IXS spectra at the (**a–d**) *M* and (**e–h**) *L* points at (**a,e**) 302 K, (**b,f**) 197 K, (**c,g**) 127 K, and (**d,h**) 102 K showing individual phonon modes, the elastic line and the overall sum of all of these contributions (full fit). **i,j** The fitted undamped

phonon energies (see Methods) as a function of temperature for (**i**) the *M* point (**j**) the *L* point alongside calculated anharmonic values for the modes which show softening with temperature (stars). The error bars are obtained from the covariance matrix of the fit and therefore already include the propagated uncertainties of both the fitted energy and linewidths of the modes. The lines represent guides to the eye.

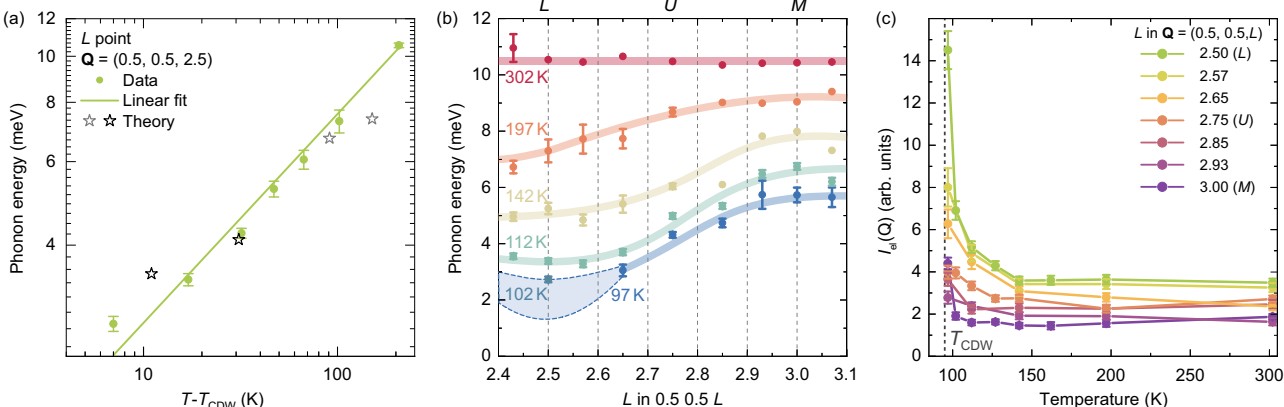

**Fig. 5 | Order of the transition, scale of the softening effect in reciprocal space and growth of the elastic peak. a** A log-log plot of the energy of Phonon B at the *L* point as a function of the reduced temperature $T - T_{CDW}$, along with calculated values for the softening mode before and after the avoided crossing (stars). **b** The energy of Phonon B along the high symmetry line $\mathbf{Q} = (0.5, 0.5, L)$ as a function of temperature. The lines are guides to the eye. Some points are omitted at 97 K due to

the phonon approaching overdamping, with data from 102 K being shown instead for the *L* point (triangle). The shaded region indicates approximate boundaries for the energy at 97 K for these points. **c** The intensity of the fitted elastic peak as a function of temperature along the line $\mathbf{Q} = (0.5, 0.5, L)$. In all panels, the error bars are obtained from the covariance matrix of the fit.

Fig. 4j and Supplementary Information) and transfer of spectral weight between two $L_2^-$ modes at about 5 meV. The avoided crossing is absent by symmetry at *M*. Due to the overlapping phonons in this energy region and the experimental resolution, however, fitting cannot capture the anti-crossing, most pronounced in the vicinity of *L*.

In order to examine the temperature dependence of the energy of the soft branch in more detail, Fig. 5a shows the log-log energy dependence of this branch as a function of the relative temperature $T - T_{CDW}$ for both the experimental data and the anharmonic calculations. There is a good agreement between the data and a linear fit where the fitted gradient is $0.44 \pm 0.02$. Due to the proximity of this value to 0.5, it is impossible for us to completely rule out that the CDW transition shows mean field, second-order-like behavior from this measurement alone, but we note that our measurements of $T_{CDW}$ from the zero-energy intensity, shown in Supplementary Fig. 4, do show a

small hysteresis ( ~ 1 K), in line with the weakly first-order behavior observed in thermodynamic measurements[7,47,53], also supported by free energy calculations including lattice anharmonicity[37].

Having established the existence of a phonon anomaly at the *M* and *L* points, we now discuss the dispersion of this phonon along the *M-L* line $\mathbf{Q} = (0.5, 0.5, L)$, as shown in Fig. 5b. At 302 K, the phonon is essentially non-dispersive, with a momentum-independent energy of around 10.3 meV. Even at 197 K, however, there is a clear dispersive effect: not only has the phonon softened everywhere along this line, but the degree of softening increases essentially monotonically from the *M* to the *L* point. As the temperature decreases towards $T_{CDW}$, the phonon continues to soften along the whole line, but the dispersive effect remains and strengthens. This effect confirms that the *L* point is the leading lattice instability in the system but also reveals that the phonon anomaly is unusually broad in reciprocal space. Additionally,

in order to gauge the size of the anomaly in-plane, we also performed measurements from the $M$ point $\mathbf{Q} = (0.5, 0.5, 3)$ towards the $\Gamma$ point $\mathbf{Q} = (1, 0, 3)$, which reveal that the softening effect gradually decreases away from $M$ and fully disappears by $\mathbf{Q} = (0.75, 0.25, 3)$, halfway between $\Gamma$ and $M$, as detailed in Supplementary Fig. 5.

As is clear even from the raw spectra in Fig. 3, with decreasing temperature, as well as the growing contribution to the low-energy spectral weight due to the softening phonon, the elastic scattering is also enhanced. Figure 5c shows the fitted elastic scattering intensity, $I_{el}(Q)$, as a function of temperature for $Q$-points along the line between $M$ and $L$ (see TDS maps in Supplementary Note 10). The intensity increases for all points with decreasing temperature, but most strongly in proximity to $L$. Thermodynamic, transport, and diffraction measurements show a sharp CDW transition (see Supplementary Fig. 4)[47], ruling out a sizable distribution of local transition temperatures as the origin of this elastic precursor. Therefore, rather than the ordering of a small fraction of the sample at higher temperature, this enhancement of the elastic line signals the development of intrinsic, slowly fluctuating CDW correlations. As the transition is approached, these fluctuations slow down and grow in spatial extent, giving rise to resolution-limited superlattice Bragg peaks below $T_{CDW}$. This in turn indicates that, in addition to the displacive, soft-mode character heralded by the pronounced softening of the $L$ point phonon, the CDW transition contains also an order-disorder, central-peak-like component, consistent with ref. 45.

Taken together, our experimental and theoretical results demonstrate that the CDW is driven by the softening of an $L_2^-$ phonon, identifying the $L$ point as the leading instability and favoring a three-dimensional $2 \times 2 \times 2$ ordering pattern. In general, the phonon self-energy that governs such instabilities contains both the electronic phase-space factor and the electron-phonon matrix elements, so that the commonly used distinction between Fermi-surface-nesting-driven and electron-phonon-coupling-driven CDWs should be understood in an operational sense. In this framework, a nesting-driven instability would correspond to a sharp enhancement of the electronic phase space, whereas an EPC-driven instability arises when the momentum dependence of the electron-phonon matrix elements dominates. The phonon softening effect reported here is broad in reciprocal space: the phonon softens by at least 4 meV everywhere along the $M$-$L$ line and the softening continues to halfway between $\Gamma$ and $M$ in-plane, encompassing half the BZ and closely matching the region where calculations predict strong EPC in the unstable branch (Supplementary Note 12). This is inconsistent with a sharp Kohn anomaly driven by a nesting of the van Hove singularities at the $M$ point[3,29–34,38], and rather suggests a correlation driven effect, more similar to established anisotropic EPC-driven CDW materials such as $2H - NbSe_2$, $TbTe_3$, and $1T - VSe_2$[54–56] than systems which host a Kohn anomaly[57–59]. The EPC driven CDW is also consistent with first-principles calculations that do not see any correlation of the nesting with the phonon softening[38].

This conclusion is also in agreement with hydrostatic pressure studies, which found a high sensitivity of the CDW wavevector and ordering temperature to pressure, even at pressures below 1 GPa, despite calculations based on the high pressure structure suggesting there is little change to the van Hove singularities or the nesting[15], and a time-resolved ARPES study which found evidence for EPC being the main mechanism for the CDW formation[40]. Recent IXS measurements on $KV_3Sb_5$ have also reported soft phonons associated with CDW formation[60]. The similarity to the present $CsV_3Sb_5$ results suggests that an EPC-driven soft-phonon mechanism is likely common to the $AV_3Sb_5$ family.

We note that, in both the calculation and experimental data, the modes with the highest sensitivity are those belonging to the $M_1^+/L_2^-$ irreducible representations. Modes without this symmetry show little temperature dependence and, as shown in Supplementary Note 12,

only extremely weak electron-phonon coupling. In addition, the $M_1^+/L_2^-$ modes are most sensitive to the details of the calculation and parameters such as the electron temperature in the purely harmonic case[29,30,38].

As discussed above, the unstable phonon near the $L$ point approaches the overdamped regime at low temperature, with a fitted linewidth exceeding its damped energy. In this regime, the DHO model no longer provides a reliable description. Although a Lorentzian line shape is sometimes employed in such cases[61], the convolution with the instrumental resolution prevented us from separating these contributions here, and we therefore omitted the corresponding points from Figs. 4 and 5.

Reliable fitting close to $T_{CDW}$ is further hindered by the rapid growth of the elastic peak below 110 K, which dominates the spectra near the $L$ point and obscures the soft-phonon linewidth. By contrast, at the $U$ and $M$ points, where the elastic line is weaker, the fits are more robust and reveal that the inelastic spectra cannot be described without a substantial and continuous broadening of the soft branch (see Supplementary Information). This pronounced linewidth increase is also reproduced by first-principles calculations and originates from the combined effects of strong EPC and lattice anharmonicity. These results highlight that both play a central role in shaping the low-energy physics of kagome metals, further supporting as well the phonon origin of superconductivity in $CsV_3Sb_5$[37].

Beyond identifying the nature and microscopic origin of the CDW in $CsV_3Sb_5$, our findings establish the central role of lattice dynamics in governing the electronic instabilities of kagome metals. The strong coupling between electronic and lattice degrees of freedom demonstrated here provides a natural framework to understand how superconductivity, topology, and charge order emerge and interact in these materials. More broadly our work underscores the importance of phonon-driven instabilities in geometrically frustrated metals and paves the way for a unified understanding of correlated kagome systems, where intertwined orders emerge from the delicate balance between electronic correlations, topology and lattice anharmonicity.

## Methods
### Crystal growth
High-purity elements Cs (Alfa Aesar, 99.98%), V (Cerac/Pure, 99.9%, further purified to remove oxygen), and Sb (Gmaterials, 99.999%) were mixed in a molar ratio of 2:1:6 in an alumina crucible, sealed inside an iron container under argon atmosphere. The container was placed in a tube furnace (argon-sealed), heated to 1050 °C and held for 15 h. The melt was then cooled to 650 °C at 1.65 °C/h, at which point the furnace was tilted to decant excess flux, followed by cooling to room temperature. The crystals were washed in demineralised water to remove residual flux. The resulting single crystals were characterized prior to the IXS/TDS experiments. Their chemical composition was examined by energy-dispersive X-ray spectroscopy (EDS) using a COXEM EM-30plus electron microscope equipped with an Oxford Silicon-Drift Detector and the AZtecLiveLite software. The EDS analysis yielded an elemental composition of Cs:V:Sb = 1:3:5.1. For samples from each batch, $T_{CDW}$ and $T_c$ were verified via magnetic susceptibility measurements. Structural refinement and thermodynamic characterization of samples grown via the same protocol can be found in refs. 15 and [47].

### TDS experiment
The thermal diffuse X-ray scattering experiments were performed at beamline ID28 of the European Synchrotron Radiation Facility (ESRF)[62,63]. The incident X-ray beam energy was set to 17.794 keV and the beam was focused to a spot of approximately $40 \times 40\,\mu m^2$. The data were acquired with a Pilatus3 X 1M detector in shutterless mode while continuously rotating the sample and recording images while

integrating over an angular range of 0.25° and 0.5 s. The CrysAlis Pro software package[64] was used for determining the unit cell and sample orientation. The reciprocal space reconstructions were created with a software developed at the beamline ID28. Low-temperature conditions were achieved using an Oxford Cryostream 700 Plus cooling system.

## IXS experiments

The IXS experiments were performed at beamline ID28[62,63] of the ESRF in July 2023 and at beamline BL43LXU[65,66] of the RIKEN SPring-8 Center (Japan) in November 2024. At ID28, the incident X-ray beam energy was set to 17.794 keV using the (999) silicon reflection with an energy resolution of 3 meV. The beam was focused to a spot of $25 \times 25 \, \mu m^2$ and the momentum resolution was set to $\approx 0.25 \, nm^{-1}$ in the scattering plane and $\approx 0.75 \, nm^{-1}$ perpendicular to it. Low-temperature conditions were achieved using an Oxford Cryostream 700 Plus cooling system.

At BL43LXU, the IXS spectrometer was operated at an X-ray energy of 23.724 keV using the silicon (121212) backscattering reflection with a nominal 1.1 meV[49,50] energy resolution. A two-dimensional array of $7 \times 4 = 28$ analyzers was used to collect data from 28 different momentum transfers in each scan. The energy resolution of each analyzer was determined as discussed in ref. 67 and was 1.1–1.2 meV for most of the analyzers for which data is presented. The momentum resolution was set by $40 \times 40 \, mm^2$ slits at 9m from the sample, corresponding to a momentum resolution of $\Delta \mathbf{Q} = (\Delta H, \Delta K, \Delta L) = (0.03, 0.05, 0.06)$ reciprocal lattice units, full width. For measurements at low temperatures, the samples were mounted inside a closed-cycle cryostat.

## Dynamical structure factor calculations

In an IXS experiment, the measured intensity is determined by the product of the phonon spectral function and the one-phonon dynamical structure factor. The spectral function contains the intrinsic phonon properties, such as energy, damping, and temperature-dependent renormalization, whereas the dynamical structure factor acts as a matrix element that governs how strongly a given phonon mode contributes to the scattered intensity at a specific momentum transfer. For a phonon branch $\nu$ with wavevector $\mathbf{q}$, the one-phonon dynamical structure factor can be written as

$$S(\mathbf{Q}, \omega) \propto \sum_{\nu} \frac{1}{\omega_{\mathbf{q}\nu}} \left| \sum_{d} \frac{f_d(\mathbf{Q})}{\sqrt{M_d}} e^{-W_d(\mathbf{Q})} (\mathbf{Q} \cdot \mathbf{e}_{d,\mathbf{q}\nu}) e^{i\mathbf{Q} \cdot \mathbf{r}_d} \right|^2 \delta(\omega - \omega_{\mathbf{q}\nu}), \quad (1)$$

where $f_d(\mathbf{Q})$ is the atomic form factor of atom $d$, $M_d$ its mass, $W_d$ the Debye–Waller factor, $\mathbf{e}_{d,\mathbf{q}\nu}$ the phonon eigenvector, and $\mathbf{r}_d$ the atomic position. Because the intensity depends on the projection $\mathbf{Q} \cdot \mathbf{e}_{d,\mathbf{q}\nu}$ and interference between atomic contributions, the structure factor can vary strongly between symmetry-equivalent points in different Brillouin zones.

Using phonon eigenvectors obtained from first-principles calculations, we evaluated the dynamical structure factor across multiple experimentally accessible Brillouin zones (see Supplementary Note 5). This analysis revealed that the unstable phonon branch carries negligible intensity in several zones previously investigated, whereas the structure factor is strongly enhanced in the $\Gamma_{103}$ zone near the $M$ and $L$ points. These calculations therefore guided the choice of experimental geometry used in the present IXS measurements.

## Fitting

As described in the main text, the phonons were fitted using a damped harmonic oscillator function, $S_j(Q, E)$ which is a function of the wave vector $Q$ and the energy $E$, weighted by the Bose factor. For the $j$th phonon,

$$S_j(Q, E) = \frac{A_j}{\pi} \frac{1}{1 - e^{-\frac{E}{k_B T}}} \frac{4\Gamma_j E}{\left(E^2 - (E_j^2 + \Gamma_j^2)\right)^2 + 4(\Gamma_j E)^2}, \quad (2)$$

where $T$ is the temperature, $A_j$ is the phonon structure factor, $E_j$ the damped phonon energy of the $j$th phonon and $\Gamma_j$ is the damping rate (the HWHM of the phonon). The undamped phonon energy reported in Fig. 4 is then $E_{0j} = \sqrt{E_j^2 + \Gamma_j^2}$, which is described simply as the phonon energy in the main text. The corresponding statistical error bars $\sigma_{E_0}$ are obtained from the covariance matrix of the multi-parameter fit $\sigma_{E_0} = \sqrt{\frac{E^2 \sigma_E^2 + \Gamma^2 \sigma_\Gamma^2}{E^2 + \Gamma^2}}$, and include the propagated uncertainties $\sigma_E$ and $\sigma_\Gamma$ of both $E$ and $\Gamma$.

Due to the challenges involved in fitting the softening phonon, Phonon B, as it crosses over Phonon A and Phonon E (only observable close to at the $L$ point), the $A_j$ values for Phonons A, B and E were fixed to their 302 K values. This assumption is supported by the calculations, which see a negligibly small change in this factor as a function of temperature at the $M$ point for the softening phonon and at the $L$ point when considering the sum of the structure factor for the branches involved in the avoided crossing (here Phonons A,B).

In addition, due to the proximity of Phonons B and C at 302 K, to enable these phonons to be distinguished and not fitted as a single broad phonon, the $A_j$ of Phonon C was fixed to the fitted 197 K value during the fitting of the 302 K data—as is clear from Fig. 4, this phonon shows no significant change in energy, width or amplitude as a function of temperature, validating this procedure. Given the small intensity of the higher energy Phonon E close to the $M$ point (see Fig. 4a–d) its amplitude and width were fixed to its 302 K fitting parameters.

The resolution for each analyzer was determined at the start of the experiment by measuring tempax glass, a mostly elastic scatterer, with the residual inelastic response of the glass removed as discussed in ref. 67, generating a sarf (smooth approximation to the resolution function). The sarf, in this case a combination of four Lorentzians and three Gaussians to accurately capture both the width and the tails of the resolution function, was used in the fitting procedure.

## Computational approach

The temperature dependent quantum vibrational modes were computed within the stochastic self-consistent harmonic approximation (SSCHA) and its dynamical extension implemented in the SSCHA package[51,52,68]. Considering the diagonal terms of the phonon-phonon interaction (Bubble approximation), along with linear response electron-phonon interaction, we approximated the phonon spectral function to a sum of individual Lorentzian distributions with finite linewidth. To compare with experiments, dynamic structure calculations were included. The force and energy calculations for the SSCHA calculations were performed with Gaussian approximation potential (GAP)[69] trained on density functional theory (DFT) data with the optB88-vdW[70] functional for the exchange and correlation functional. The electron-phonon linewidth was obtained from density functional perturbation theory (DFPT)[71] as implemented in the Quantum Espresso package[72,73] making use of ultra-soft pseudopotentials[74]. Each of the steps and assumptions made in the calculation are explained in detail in the Supplementary Information.

For Fig. 1(a, c) the harmonic phonons were computed in a separate framework. IXS structure factors were calculated on the basis of ab initio phonon frequencies and eigenvectors using the density-functional perturbation theory as implemented in the mixed-basis pseudo-potential method[75]. Dynamical matrices were first calculated on a $6 \times 6 \times 2$ hexagonal mesh and then determined for arbitrary points in the Brillouin zone using standard Fourier-interpolation techniques,

from which the phonon dispersion and structure factors were derived. For this calculation, the computational parameters were the same as described in a previous publication (see supplemental material of ref. 47).

## Data availability
The experimental IXS and DS data reported in this study are available in the KIT Open repository, under the identification number https://doi.org/10.35097/2kxzs0g0n3z2v52f. All relevant computational data are available from the authors upon request.

## Code availability
All codes used in this study are open-source and available from their respective websites.

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

## Acknowledgements

We thank S. Blanco-Canosa and F. Weber for fruitful discussions. We acknowledge provision of beamtime at the RIKEN Quantum NanoDynamics Beamline, BL43LXU, under the proposal with number 20240083. We acknowledge the European Synchrotron Radiation Facility (ESRF) for provision of synchrotron radiation facilities under proposal number HC-5341.

## Author contributions

M.L.T. conceived and supervised the project. A.A.H. grew the single crystals. P.H.McG., F.H., M.J.GvW., A.K., A.B., D.I., A.Q.R.B., S.M.S. and M.L.T. carried out IXS and DS experiments. P.H.McG., F.H. and M.J.GvW. analysed the IXS and DS data. M.M. carried out XRD experiments. R.H., M.A., M.G.V. and I.E. performed first-principle calculations; R.H. carried out calculations of the harmonic phonons and M.A., M.G.V. and I.E. performed phonon calculations within the SSCHA. M.L.T., P.H.McG. and F.H. wrote the manuscript with input from all the co-authors. Open Access funding enabled and organized by Projekt DEAL.

## Funding

M.L.T., S. M. S., A.A.H., F. H., and P.H.McG. disclose support for the research of this work from the Deutsche Forschungsgemeinschaft (DFG; German Research Foundation) Project-ID 422213477-TRR 288 (Projects B03 and B10). R.H. acknowledges support by the state of Baden-Württemberg through bwHPC. The work by M.A., M.G.V. and I.E. was also supported by the PID2022-142861NA-I00 and PID2022-142008NB-I00 projects funded by MICIU/AEI/10.13039/501100011033 and FEDER, UE; the Department of Education, Universities and Research of the Eusko Jaurlaritza and the University of the Basque Country UPV/EHU (Grant No. IT1527-22); Canada Excellence Research Chairs Program for Topological Quantum Matter; NSERC Quantum Alliance France-Canada; and Diputación Foral de Gipuzkoa Programa Mujeres y Ciencia. M.A. acknowledges a PhD scholarship from the Basque Government. Open Access funding enabled and organized by Projekt DEAL.

## Competing interests

The authors declare no competing interests.
