## [Transparent Peer Review file · Nature Communications]

Soft Mode Origin of Charge Ordering in Superconducting Kagome CsV₃Sb₅

Corresponding Author: Professor Matthieu Le Tacon

Version 0:

Reviewer comments:

Reviewer #1

(Remarks to the Author)

The authors reported high-resolution inelastic X-ray scattering (IXS) results for this kagome material and showed that, at the CDW transition, the phonon at the L point is softened. High-quality experimental data is the most important part of this paper. (First-principles calculations do not match the experiment exactly, but that is okay.) I think that the paper reports important experimental data that would be of interest to researchers in the field of Kagome materials and the CDW phenomenon.

I suggest that the authors consider the following points for the general readers of Nature Communications.

1. Introduce the issue of Fermi-surface nesting versus electron-phonon coupling. The two quantities are related. The phonon self-energy is affected by $\sum_{\{mnk\}} |\langle mk+q | V_{nu}(q) | nk \rangle|^2 \delta(\epsilon_{mk+q} - \epsilon_{nk} - \omega_{qnu})$: Do the authors mean Fermi-surface-nesting-dominant by $\delta(\epsilon_{mk+q} - \epsilon_{nk} - \omega_{qnu})$ is large even if $|\langle mk+q | V_{nu}(q) | nk \rangle|$ is low, and EPC-dominant by the opposite situation? If so, then at least the problem is meaningful. But, my doubt on or objection to this dichotomy is that what people usually say one thing is FSN-dominant or EPC-dominant is in a different sense, using somewhat averaged quantities. Perhaps, it is not the fault of the authors but the fault of the first few people who shaped this field. But it would be nice if the authors investigated more thoroughly whether it is meaningful to distinguish between FSN and EPC as the origin of CDW, and whether what everyone else is doing serves the purpose.

2. IXS intensity is determined by both matrix element effects and phonon spectral function. The novelty of this work is to overcome the hurdle coming from the first part by looking at the higher Brillouin zone that previous studies did not. For the interest of the general public, please explain the two factors affecting the results of IXS and what the authors have done for both of them.

3. The error bars in Fig. 4, for example, is misleading. The error bar is for the shift, and not the broadening. However, there should also be an error bar for the broadening parameter that they used for fitting as well. Would there be some fair way of presenting the experimental data?

Reviewer #2

(Remarks to the Author)

This manuscript presents a combined experimental and theoretical investigation into the origin of the charge-density wave (CDW) in the kagome superconductor CsV₃Sb₅. Using high-resolution inelastic X-ray scattering (IXS) guided by structure factor calculations, the authors claim to identify a soft phonon mode along the M–L direction in reciprocal space, with the strongest softening occurring at the L point. First-principles calculations incorporating anharmonic effects are presented to support these observations. The study aims to resolve the long-standing debate on the CDW mechanism in this material family by establishing a phonon instability. However, there are fundamental concerns regarding the central experimental evidence. I cannot support the publication of this manuscript at Nature Communications.

Here're the major concerns:

1. The claim of the soft phonon modes, which is the central finding of this manuscript, rests entirely on the fittings of the IXS spectrums shown in Fig. 3 and 4 without clear and direct identification of the softening phonon spectral peaks. The fitted mode labeled as the softening phonon appears as an extremely broad spectral feature. In Fig. 4, particularly at 127, 102K (M

point) and 197, 127 K (L point), the purported soft mode is barely distinguishable from the background, raising serious concern about whether it represents a distinct phonon branch rather than background or just a fitting artifact. Thus, with close inspection of the IXS data, the evidence for the softening mode is not convincing.

2. The authors argue that previous IXS study [Subires et al. Nat. Commun. 14, 1015 (2023)] failed to observe phonon anomalies due to unfavorable structure factors in the measured Brillouin zones. However, the Subires et al. study specifically examined the L point at $Q=(3.5, 2.5, 0.5)$ with high energy resolution (1.5 meV) and found no softening. While the present manuscript argues that this particular L point has a vanishing structure factor for the unstable branch, the claim that the L point in the Γ_{103} zone exhibits dramatic softening should be supported by a direct comparison of the IXS spectra between the two zones. Such an experimental comparison under the same sample and measurement condition can verify whether the difference shown in the calculation indeed make the soft mode detectable, and if the soft mode is as pronounced as claimed, it should be apparent by comparing the raw and unfitted data.

3. The manuscript emphasizes the substantial broadening of the soft phonon arising from electron-phonon coupling and anharmonicity. However, the linewidths extracted from the IXS fitting reveal inconsistent trends. At the L point, the linewidths maintain substantially small at low temperatures with only small fitting errors. This discrepancy further highlights the challenges and uncertainties inherent in the spectral fitting of the IXS data, raising doubts against the reliability of the extracted soft mode behavior.

Two more minor issues. The claim that "fewer than 10% of BZs exhibit an L point structure factor for this branch that reaches even half the intensity calculated at $Q=(0.5, 0.5, 2.5)$ " is presented without supporting data. Evidence for this statistic should be provided. Additionally, the manuscript contains several incomplete or erroneous references to the SI, such as "see Supplementary Section ??".

Reviewer #3

(Remarks to the Author)

This work reports the discovery of soft phonons that lead to the CDW in CsV₃Sb₅, and showed why they were not observed previously, namely due to the structure factor being too small in some zones. The absence of soft phonons is often used to argue in favor of an unconventional nature of the CDW in these topical kagome materials, and several theoretical works were presented to explain their absence. In the context of these works, the findings in this work are quite significant and could thus be published in Nature Communications.

However, there are also several issues that the authors need to address.

The main issue is that the soft phonons, and particularly their temperature evolution, are not so obvious in the raw data, although they become clearer in the detailed analysis in Figure 3. I think the data is reasonably convincing that there are soft phonons, but am not sure about how reliable the fit parameters are. Are the small error bars for the phonon energies reliable? How do they change if some fixed parameters are allowed to vary freely? It would be helpful if the authors can present in a single plot, presumably in the SI, the temperature dependence of all the parameters in the model, and clarify which are fixed and which are free. A brief discussion of how robust the soft phonon fit parameters are, when more parameters are allowed to vary, would also be helpful.

Minor issues:

1. Quantum anharmonic effects are suggested to play a pivotal role, why is this the case?
2. Fig. 1 shows realistic phonon broadenings due to anharmonicity and electron phonon coupling, but these are difficult to see in the color plot. Are these significant for understanding this work, and how do they compare with experimental results (fitted damping values)?
3. The 97 K data is suggested to become overdamped, but this means the phonon energy suddenly goes from 3meV or so to 0 meV within 5 K. Can the authors comment on how reliable this claim is, or perhaps soften the claim?
4. The connection between theory and data should be better explained. For $L=0.5$, two modes are expected in theory, but only one is seen experimentally, which mode does it correspond to in the calculations, and why is the other mode not seen?

Version 1:

Reviewer comments:

Reviewer #1

(Remarks to the Author)

The authors have successfully addressed the issues I raised. I recommend publication of the paper in Nature Communications.

Reviewer #2

(Remarks to the Author)

In the response letter and the revised manuscript, the authors have addressed and clarified the concerns and doubts that I raised in detail. In particular, the Bose-factor corrected spectra shown in Figure 3 now show the phonon mode evolution directly. Thus, I support the publication of this manuscript on Nature Communications.

Reviewer #3

(Remarks to the Author)

The authors made significant efforts to improve the manuscript, the inclusion of Bose corrected raw data makes it clear that there are certainly temperature dependent changes to the phonons. Given the numerous works that report unconventional properties of the CDW that are interpreted in terms of nesting between van Hove points, the electron-phonon mechanism presented in this work is significant, and makes this work suitable for Nat. Commun.

There are still a few minor issues that need to be addressed, before the work can be accepted.

1. For the error bars, were they determined when all the free parameters were allowed to vary? Sometimes people (incorrectly) fix 'free' parameters when calculating the error bars, which results in very small error bars.
2. For the growth of the elastic peak, is this intrinsic, or possibly arises from a distribution of transition temperatures? Even an extremely small fraction of the sample with an elevated T_{cdw} could give a large elastic signal slight above the mean T_{cdw} . If this is intrinsic, then there is both order-disorder and phonon softening processes, which makes this work consistent with Ref. 45.
3. Soft phonons were also reported in $KV3Sb5$ recently, are these similar to what is seen here in $CsV3Sb5$? If the soft phonons behave differently in the two systems, could these differences be related to the more complex phases and behaviors in $CsV3Sb5$, such as the $2 \times 2 \times 4$ phase, nematicity, and chiral transport?
4. There are many '??' in referencing the SI.

Reviewer #1 Remarks to the Author:

The authors reported high-resolution inelastic X-ray scattering (IXS) results for this kagome material and showed that, at the CDW transition, the phonon at the L point is softened. High-quality experimental data is the most important part of this paper. (First-principles calculations do not match the experiment exactly, but that is okay.) I think that the paper reports important experimental data that would be of interest to researchers in the field of Kagome materials and the CDW phenomenon.

I suggest that the authors consider the following points for the general readers of Nature Communications.

1. Introduce the issue of Fermi-surface nesting versus electron-phonon coupling. The two quantities are related. The phonon self-energy is affected by $\sum_{\{m\mathbf{k}\}} |\langle m\mathbf{k}+q | V_{\nu}(q) | n\mathbf{k} \rangle|^2 \delta(\epsilon_{m\mathbf{k}+q} - \epsilon_{n\mathbf{k}} - \omega_{\nu})$: Do the authors mean Fermi-surface-nesting-dominant by $\delta(\epsilon_{m\mathbf{k}+q} - \epsilon_{n\mathbf{k}} - \omega_{\nu})$ is large even if $|\langle m\mathbf{k}+q | V_{\nu}(q) | n\mathbf{k} \rangle|$ is low, and EPC-dominant by the opposite situation? If so, then at least the problem is meaningful. But, my doubt or objection to this dichotomy is that what people usually say one thing is FSN-dominant or EPC-dominant is in a different sense, using somewhat averaged quantities. Perhaps, it is not the fault of the authors but the fault of the first few people who shaped this field. But it would be nice if the authors investigated more thoroughly whether it is meaningful to distinguish between FSN and EPC as the origin of CDW, and whether what everyone else is doing serves the purpose.

Our Answer: We thank the reviewer for this thoughtful comment regarding the distinction between FSN and EPC as driving mechanisms CDW formation.

As the reviewer correctly points out, the phonon self-energy formally contains both the electronic phase space factor (represented by the energy-conserving δ -function) and the electron–phonon matrix elements. In practice, however, one rarely analyzes these terms directly at the level of individual electronic states of momentum k . Instead, it is customary to consider momentum-averaged quantities such as the phonon linewidth given by the electron-phonon interaction, the electronic joint density of states that is usually referred as the nesting function, or ratios between these quantities, which provide a more practical way of separating the influence of electronic phase space and matrix elements.

This issue has been discussed in detail in our previous work on BaNi_2As_2 (see Fig. 4 in PRL 129, 247602 (2022) and Figs. 10–11 of the Supplementary Information of PRB 108, 224115 (2023)). In those studies, we explicitly compared the contributions arising from the electronic phase space and from the EPC matrix elements and showed that the momentum dependence of the EPC matrix elements plays a dominant role.

More recently, in Comm. Materials 5, 234 (2024), we performed a more explicit comparison of these contributions in the case of CsV_3Sb_5 and demonstrated that the observed phonon anomalies are primarily governed by the momentum dependence of the EPC rather than by FSN effects alone. In fact, the phonon softening obtained at the harmonic level correlates well with the phonon linewidth, which contains both the contribution of the electron-phonon matrix elements and the electronic scattering phase space, but not at all with the nesting function. This is a clear sign that the instability is driven by the EPC.

In the present work, we therefore adopt the commonly used terminology of “nesting-driven” versus “EPC-driven” instabilities in this practical sense: a nesting-driven instability would primarily reflect a strong peak in the electronic phase space factor, whereas an EPC-driven instability arises when the momentum dependence of the electron–phonon matrix elements plays the dominant role. The unusually broad momentum extent of the phonon softening observed here, extending over a large fraction of the Brillouin zone, is inconsistent with a sharp nesting-driven Kohn anomaly and instead closely matches the momentum dependence expected from strong EPC, in agreement with our previous theoretical calculations.

To clarify this conceptual point for the reader, we have edited the beginning of discussion in the revised manuscript.

2. IXS intensity is determined by both matrix element effects and phonon spectral function. The novelty of this work is to overcome the hurdle coming from the first part by looking at the higher Brillouin zone that previous studies did not. For the interest of the general public, please explain the two factors affecting the results of IXS and what the authors have done for both of them.

Our Answer: We thank the reviewer for this helpful suggestion and agree that the distinction between matrix-element effects and the phonon spectral function is important for readers who may not be familiar with IXS. In an IXS experiment the measured intensity is determined by two main factors:

- (i) the phonon spectral function, which contains the intrinsic phonon properties such as energy, damping, and temperature-dependent renormalization, and
- (ii) the dynamical structure factor (matrix element), which governs how strongly a given phonon mode contributes to the scattered intensity at a specific momentum transfer.

While the spectral function reflects the intrinsic lattice dynamics, the dynamical structure factor depends sensitively on the scattering geometry and varies strongly between symmetry-equivalent points in different Brillouin zones. As a result, a phonon that is strongly renormalized may nevertheless be essentially invisible in certain zones if the corresponding structure factor is small.

In the present work, we explicitly calculated the dynamical structure factor for the unstable phonon branch across multiple Brillouin zones in order to identify scattering geometries where the mode carries substantial intensity. This guided the choice of the Γ_{103} zone, where the structure factor of the unstable branch is particularly favorable, allowing the phonon anomaly to be detected experimentally. In contrast, several zones previously investigated have a vanishingly small structure factor for this mode, which naturally explains why no anomaly was reported there.

To clarify this point for a broader audience, we have added a dedicated ‘Dynamical structure factor calculations’ section in the methods sections in the manuscript and the Supplementary Information to more explicitly explain the respective roles of the phonon spectral function and the dynamical structure factor in determining the measured IXS intensity.

3. The error bars in Fig. 4, for example, is misleading. The error bar is for the shift, and not the broadening. However, there should also be an error bar for the broadening parameter that they used for fitting as well. Would there be some fair way of presenting the experimental data?

Our Answer: We thank the reviewer for this comment and for the opportunity to clarify the presentation of the uncertainties.

The error bars shown in Fig. 4 correspond to the uncertainties on the extracted undamped phonon frequencies obtained from the fits. These uncertainties already incorporate the correlations between the fitted parameters, including the linewidth, since the undamped phonon energy $E_0 = \sqrt{E^2 + \Gamma^2}$ depends explicitly on the fitted Γ . The error bars shown for E_0 are obtained from the covariance matrix of the fit and therefore already include the propagated uncertainties of both E and Γ .

We chose to present Γ separately in the Supplementary Information because, as discussed above, their extraction is more sensitive to the details of the fitting when phonon modes overlap or approach the overdamped regime. For this reason, and since the main focus of the manuscript is the temperature evolution of the phonon energy, we considered it clearer to present this analysis separately from the main figure.

To address the reviewer's concern, we have clarified the definition of the error bars in the caption of Fig. 4 in the revised manuscript and added a discussion regarding the uncertainties of the fitted parameters in the Supplementary Information.

Reviewer #2 Remarks to the Author:

This manuscript presents a combined experimental and theoretical investigation into the origin of the charge-density wave (CDW) in the kagome superconductor CsV₃Sb₅. Using high-resolution inelastic X-ray scattering (IXS) guided by structure factor calculations, the authors claim to identify a soft phonon mode along the M–L direction in reciprocal space, with the strongest softening occurring at the L point. First-principles calculations incorporating anharmonic effects are presented to support these observations. The study aims to resolve the long-standing debate on the CDW mechanism in this material family by establishing a phonon instability. However, there are fundamental concerns regarding the central experimental evidence. I cannot support the publication of this manuscript at Nature Communications.

Our Answer: While we respectfully disagree with the reviewer's assessment, we appreciate the careful reading of our manuscript and the opportunity to clarify and strengthen the presentation of our experimental evidence. Below, we address each concern in detail and provide additional analyses to make the identification of the soft phonon mode fully transparent and robust.

Here're the major concerns:

1. The claim of the soft phonon modes, which is the central finding of this manuscript, rests entirely on the fittings of the IXS spectrums shown in Fig. 3 and 4 without clear and direct identification of the softening phonon spectral peaks. The fitted mode labeled as the softening phonon appears as an extremely broad spectral feature. In Fig. 4, particularly at 127, 102K (M point) and 197, 127 K (L point), the purported soft mode is barely distinguishable from the background, raising serious concern about whether it represents a distinct phonon branch rather than background or just a fitting artifact. Thus, with close inspection of the IXS data, the evidence for the softening mode is not convincing.

Our Answer: We thank the reviewer for this important comment regarding the identification of the soft phonon mode.

We respectfully disagree with the statement that the claim of softening relies entirely on fitting. The temperature-dependent anomaly can, in fact, be directly identified in the raw IXS data once the fundamental characteristics of IXS spectra are properly taken into account.

First, it is important to clarify that IXS spectra do not contain an arbitrary background in the usual sense. The measured intensity consists of a resolution-limited elastic line and of inelastic contributions from phonon excitations.

The phonon intensities exhibit a strong temperature dependence governed by Bose-Einstein statistics, whereas the elastic line is, in high-quality samples and well-controlled experiments, only weakly (if at all) temperature dependent, except at reciprocal-space locations where quasi-static charge correlations develop, as is the case near the CDW wavevector (and in fact along the entire M-L reciprocal line) in this material.

A key experimental advantage of the BL43LXU setup at SPring-8 is that 28 analyzers simultaneously record spectra at different momentum transfers. When the reference analyzer (analyzer A06) is centered on the L point of interest [here at (0.5, 0.5, 2.5) – the question of why this point and not any arbitrary L-point is answered next], the surrounding analyzers probe nearby Q positions under identical experimental conditions.

This provides an internal consistency check that is independent of any fitting procedure.

Inspection of the raw spectra collected between 302 K and 97 K (Figure 1) shows:

- Away from the M-L line, the resolution-limited elastic line exhibits minimal temperature dependence, and the phonon spectra evolve exactly as expected from the Bose factor (as shown below).
- At the L point (A06 in Figure 1), in contrast, a clear additional low-energy enhancement develops upon cooling.

Figure 1: Raw IXS data obtained at 28 different Q-points on beamline BL43LXU at SPring-8 at temperatures of 302, 197, 162, 142, 102 and 97K. In this plot, the A06 analyzer is centered at the L point (0.5, 0.5, 2.5). The cyan-colored lines are fits of the elastic lines of the 302 K data.

To isolate intrinsic phonon behavior, we proceed in two transparent and model-independent steps already at the data collection step (and prior to any fitting procedure), as illustrated by Figure 2 shown in this reply:

1. Subtract the measured resolution-limited elastic contribution.
2. Apply Bose-factor correction to the inelastic intensity.

The resulting Bose-corrected spectra (Figure 2) demonstrate unambiguously that:

- At randomly sampled Q points away from the M-L line, the inelastic spectra at different temperatures collapse onto one another.
- At the L point (and along the M-L direction), a pronounced redistribution of spectral weight toward low energies develops upon cooling.

Figure 2: Same IXS dataset as in the previous figure, after subtraction of the elastic line and Bose-correction.

As show in Figure 2 of this reply, this spectral-weight redistribution is directly visible in the processed raw data and does not rely on multi-parameter fitting. We have redesigned Fig. 3 to make this clear. In particular we have added two panels comparing the Bose-factor-processed raw data along the M-L line (measured on analyzer 6) and on different sets of Q-points (measured simultaneously on analyzer A38).

We have also added a supplementary figure (S3), which is reproduced below, comparing the data at the M and L points as a function of temperature with simulations of the low temperature spectra assuming that the room-temperature spectra evolve upon cooling solely according to the Bose factor (with a temperature-independent elastic line). The comparison clearly shows that the observed temperature dependence is substantial and cannot be explained by trivial Bose-factor effects alone, independently of any fitting procedure.

Figure 3 Comparison between experimental spectra measured at the M and L points at various temperatures and spectra simulated assuming only a trivial Bose-factor temperature dependence inferred from the room-temperature data. The deviations clearly indicate a redistribution of spectral weight beyond the Bose-factor effect.

The fitting procedure serves to quantify the magnitude of the renormalization, but we insist that the qualitative anomaly is already evident **prior to any fitting**. Furthermore, the absence of comparable temperature-dependent changes at other simultaneously measured Q points demonstrates that the observed effect is not an experimental or fitting artifact.

2. The authors argue that previous IXS study [Subires et al. Nat. Commun. 14, 1015 (2023)] failed to observe phonon anomalies due to unfavorable structure factors in the measured Brillouin zones. However, the Subires et al. study specifically examined the L point at $Q=(3.5, 2.5, 0.5)$ with high energy resolution (1.5 meV) and found no softening. While the present manuscript argues that this particular L point has a vanishing structure factor for the unstable branch, the claim that the L point in the Γ_{103} zone exhibits dramatic softening should be supported by a direct comparison of the IXS spectra between the two zones. Such an experimental comparison under the same sample and measurement condition can verify whether the difference shown in the calculation indeed make the soft mode detectable, and if the soft mode is as pronounced as claimed, it should be apparent by comparing the raw and unfitted data.

Our Answer: We thank the reviewer for raising this important point regarding the choice of Brillouin zone and its impact on the visibility of the soft phonon.

In the study by Subires et al., the selection of the measured L point appears to have been guided primarily by diffuse scattering measurements. This is a legitimate and commonly used strategy. However, diffuse scattering intensity can be dominated by elastic and quasi-elastic contributions and does not necessarily imply that the inelastic spectral weight of a specific phonon branch will

be sizeable at that same Q point. In the case of CVS, clearly, the elastic contribution appears to be the main component of the diffuse intensity.

As demonstrated in our Fig. 1, the dynamical structure factor of the unstable L_2^- branch varies strongly between L points in different Brillouin zones. For the specific L point investigated by Subires et al. ($Q = (3.5, 2.5, 0.5)$, i.e. L-point of the Γ_{420} zone), the calculated structure factor of the unstable branch is negligibly small. Therefore, even with high energy resolution, the soft phonon would be essentially invisible in that geometry. This is fully consistent with the absence of any reported anomaly at that location.

More generally, within the accessible momentum space at our working photon energy ($2\theta < 50^\circ$), on the order of $\sim 10^4$ crystallographically equivalent L points are in principle reachable. Our structure factor calculations at the M and L points show that only a very small fraction of these exhibit appreciable intensity for the unstable branch (see detailed quantitative analysis added to the Supplementary Information – a note regarding structure factor calculation has also been added to the Methods section). In most zones, the structure factor is vanishingly small, rendering detection of the anomalous phonon experimentally impractical without prior calculation.

Structure factor calculations are a standard and well-established prerequisite for inelastic scattering experiments, precisely because phonon visibility depends critically on matrix element effects. Our approach therefore does not contradict the previous work but rather clarifies why the soft mode was not observable in the specific Brillouin zone previously explored.

3. The manuscript emphasizes the substantial broadening of the soft phonon arising from electron-phonon coupling and anharmonicity. However, the linewidths extracted from the IXS fitting reveal inconsistent trends. At the L point, the linewidths maintain substantially small at low temperatures with only small fitting errors. This discrepancy further highlights the challenges and uncertainties inherent in the spectral fitting of the IXS data, raising doubts against the reliability of the extracted soft mode behavior.

Our Answer: We agree with the reviewer that extracting reliable linewidths in the presence of overlapping modes and strong renormalization can be experimentally challenging, and we have taken particular care to assess the robustness of the fitted parameters.

A central observation of our data is that, once corrected for the Bose factor, the IXS spectra away from the M-L line are essentially temperature independent. In contrast, the spectra along the M-L line exhibit a pronounced redistribution of spectral weight as the temperature approaches the CDW transition. This behavior is directly visible in the data and constitutes the central experimental result of this work, independent of the details of the fitting procedure.

Even along the M-L line several phonons show little or no temperature dependence. This can be seen in the new Fig. 3f comparing Bose-corrected spectra at 97 K and 302 K. The stability of these modes (and of most phonons recorded simultaneously on the 28 analyzers) provides a natural constraint for the analysis and forms the basis of our fitting procedure. In particular, we keep the oscillator strengths of the phonons A and B (the soft one) fixed to their room-temperature values while allowing the energy and width parameters of the soft mode to evolve. As the soft mode energy decreases, its spectral weight naturally increases as $1/E$ within the DHO model. The enhanced low-energy intensity observed upon cooling is therefore quantitatively reproduced without changing the oscillator strength, by varying only the phonon energy and width. The resolution-limited elastic line and the high experimental resolution further help constrain the fits. The small error bars on the linewidth reflect the stability of the fit under these constraints.

One temperature point (162 K) shows larger uncertainty because the maximum intensity of the soft mode lies close to that of a strong phonon near 5 meV (note that the undamped phonon frequency shown in Fig. 4d differs from the peak position, as explained in the manuscript).

In the intermediate temperature range (≈ 140 – 180 K), the spectral weight of the anomalous branch is distributed across two modes. As shown by the calculations, these two L_2^- modes undergo an avoided crossing and exchange spectral weight. Because both branches carry significant structure factor, their spectral overlap leads to an apparently larger effective linewidth when modeled with independent damped harmonic oscillators. The increased fitted width in this range is therefore qualitatively consistent with the theoretical picture of hybridization and spectral weight redistribution discussed in the supplementary material (see Fig. S2).

At lower temperatures (below ~ 120 K), the situation changes qualitatively: the upper branch loses most of its structure factor while the lower branch continues to soften. Once the soft branch dominates the spectral weight, the fitting becomes more stable as spectral overlap is reduced. In this regime the linewidth extracted from the DHO model should not be interpreted as a direct measure of the phonon lifetime; rather, the mode approaches an overdamped regime. Close to T_{CDW} , the growing elastic line also prevents a reliable extraction of the phonon energy.

Importantly, the key experimental observation—the strong redistribution of spectral weight toward low energies upon cooling—does not depend on the precise numerical value of the fitted linewidth. This behavior is directly visible in the Bose-corrected spectra and remains robust against reasonable variations of the fitting constraints.

To clarify these points, we included a discussion on the fitting along the lines mentioned above in the supplementary information.

We therefore conclude that the linewidth behavior is consistent with the combined effects of anharmonicity, mode hybridization, and strong electron–phonon coupling, and does not undermine the identification of the soft phonon instability.

Two more minor issues. The claim that "fewer than 10% of BZs exhibit an L point structure factor for this branch that reaches even half the intensity calculated at $Q=(0.5, 0.5, 2.5)$ " is presented without supporting data. Evidence for this statistic should be provided.

Our Answer: as specified above, within the accessible momentum space at our working photon energy ($2\theta < 50^\circ$), on the order of $\sim 10^4$ crystallographically equivalent L points are in principle reachable. We have reduced this number to about $\sim 10^3$, most of which are not even reasonably accessible considering the morphology of the samples and the geometrical constraints of the sample environment.

We now present the results of the calculation for the structure factor of the unstable mode at M and L points for Brillouin zones with $L < 4$ in the supplementary information. It provides direct illustration that the actual number of zones in the soft unstable mode has sizeable intensity both at M and L points is very limited and that the chosen 103 zone is the optimal one.

Additionally, the manuscript contains several incomplete or erroneous references to the SI, such as "see Supplementary Section ??".

Our Answer: we thank the reviewer for spotting this out and corrected this and a few other typos.

Reviewer #3 Remarks to the Author:

This work reports the discovery of soft phonons that lead to the CDW in CsV₃Sb₅, and showed why they were not observed previously, namely due to the structure factor being too small in some zones. The absence of soft phonons is often used to argue in favor of an unconventional nature of the CDW in these topological kagome materials, and several theoretical works were presented to explain their absence. In the context of these works, the findings in this work are quite significant and could thus be published in Nature Communications. However, there are also several issues that the authors need to address.

Our Answer: we thank the reviewer for their positive assessment of our work and address the issues raised in their report hereafter.

The main issue is that the soft phonons, and particularly their temperature evolution, are not so obvious in the raw data, although they become clearer in the detailed analysis in Figure 3.

Our Answer: We thank the reviewer for this comment. A similar concern regarding the visibility of the soft phonon in the raw data was also raised by Reviewer #2, which motivated us to improve the presentation of the data in the revised manuscript.

In particular, we have redesigned Fig. 3 to more clearly illustrate the evolution of the spectra. As discussed in the text, the apparent temperature dependence of the raw IXS spectra is dominated by the Bose factor, which introduces a strong but trivial temperature dependence of the phonon intensities. Once this effect is removed by applying the Bose correction, the intrinsic spectral changes become much clearer.

After this correction, the phonon softening along the M-L line is directly visible in the data as a pronounced redistribution of spectral weight toward lower energies upon cooling. The detailed quantitative analysis of this effect is presented in Fig. 4, while the revised Fig. 3 (and S6) now explicitly shows the intermediate steps that make this behavior apparent at the level of the spectra.

I think the data is reasonably convincing that there are soft phonons, but am not sure about how reliable the fit parameters are. Are the small error bars for the phonon energies reliable? How do they change if some fixed parameters are allowed to vary freely? It would be helpful if the authors can present in a single plot, presumably in the SI, the temperature dependence of all the parameters in the model, and clarify which are fixed and which are free. A brief discussion of how robust the soft phonon fit parameters are, when more parameters are allowed to vary, would also be helpful.

Our Answer: We thank the reviewer for this thoughtful remark regarding the robustness of the fitted parameters.

The fitting procedure is based on the observation that most phonons exhibit no intrinsic temperature dependence once the Bose factor is considered. In other words, their characteristic frequency, linewidth, and oscillator strength remain essentially constant with temperature. This behavior is directly visible in the Bose-corrected spectra and provides a physically motivated constraint for the fitting procedure.

The main constraint was to keep the oscillator strengths of the A and B phonons fixed to their room-temperature values while allowing the energy and width parameters of the soft mode (B) to evolve. As the soft mode energy decreases, its spectral weight naturally increases as $1/E$ within

the DHO model. The enhanced low-energy intensity observed upon cooling is therefore quantitatively reproduced without changing the oscillator strength, by varying only the phonon energy and width. The resolution-limited elastic line and the high experimental resolution further help constrain the fits. The small error bars on the linewidth reflect the stability of the fit under these constraints.

We emphasize that even with these constraints, some of the fits are not trivial, particularly at the L point and in the intermediate temperature range where the soft mode overlaps with nearby phonons (see reply to reviewer #2). However, without including the soft mode, acceptable fits to individual spectra can simply not be obtained. This is also evident from new Fig. S3 which comparing the temperature dependence of the spectra at M and L point with the simulations obtained assuming simple Bose factor driven evolution of the spectra.

Importantly, the central experimental observation, the redistribution of spectral weight toward low energy along the M-L line upon cooling, is directly visible in the Bose-corrected spectra and does not depend on the precise numerical values of the fitted parameters. The fitting procedure therefore serves primarily to quantify the softening, while the qualitative behavior is evident from the data itself.

We have added more details about the fitting procedure in the supplementary information.

Minor issues:

1. Quantum anharmonic effects are suggested to play a pivotal role, why is this the case?

Our Answer: We thank the reviewer for this comment and for the opportunity to clarify this point.

Within the harmonic approximation, the phonon calculations predict a strong lattice instability with imaginary phonon frequencies along the M-L direction. Taken at face value, such a harmonic instability would imply that the high-symmetry phase is unstable already at very high temperature, which is inconsistent with the experimentally observed transition temperature T_{CDW} . Anharmonic effects stabilize the high-temperature structure and renormalize the phonon spectrum as a function of temperature. When these effects are included through the SSCHA method, the unstable harmonic branch is dynamically stabilized at high temperature but softens progressively upon cooling, reproducing both the observed phonon renormalization and the correct transition temperature.

In kagome superconductors this anharmonic renormalization is particularly important because the lattice contributes significantly to the free-energy balance at the CDW transition. In other words, the stabilization of the high-temperature phase is largely driven by lattice entropy, which cannot be captured within the harmonic approximation.

2. Fig. 1 shows realistic phonon broadenings due to anharmonicity and electron phonon coupling, but these are difficult to see in the color plot. Are these significant for understanding this work, and how do they compare with experimental results (fitted damping values)?

Our Answer: We thank the reviewer for this comment. The linewidths shown in Fig. 1 originate from the calculated phonon self-energies including both anharmonic phonon-phonon interactions and electron-phonon coupling. In the color representation of the figure these broadenings are indeed difficult to discern because the main purpose of Fig. 1 is to highlight the dispersion and dynamical structure factor of the unstable branch, which guided the choice of the experimental scattering geometry.

The calculated broadenings are nevertheless physically meaningful. In particular, they indicate that the unstable branch is expected to experience substantial damping as the transition is approached, reflecting the combined effects of strong electron–phonon coupling and anharmonicity. This behavior is consistent with the experimental observation that the soft mode becomes strongly renormalized and eventually overdamped close to T_{CDW} .

A detailed quantitative comparison between calculated and fitted linewidths is challenging, both experimentally and theoretically. Experimentally, the extraction of linewidths is complicated by spectral overlap between nearby phonon branches and by the limited experimental resolution. Theoretically, the calculated linewidths depend sensitively on the details of the anharmonic treatment and the electronic temperature used in the simulations. For these reasons we focus the comparison primarily on the energy renormalization and temperature evolution of the unstable branch, which are more robustly determined and show good qualitative agreement between theory and experiment.

3. The 97 K data is suggested to become overdamped, but this means the phonon energy suddenly goes from 3meV or so to 0 meV within 5 K. Can the authors comment on how reliable this claim is, or perhaps soften the claim?

Our Answer: We thank the reviewer for raising this point. We agree that once the phonon becomes strongly damped close to T_{CDW} , extracting a well-defined phonon energy becomes increasingly unreliable.

In the vicinity of the transition the spectral response evolves toward an overdamped regime, where the phonon linewidth becomes comparable to or larger than its energy. In this situation the damped harmonic oscillator parametrization used for the fits no longer provides a unique or physically transparent description of the excitation, and the extracted “energy” should not be interpreted as the true eigenfrequency of the mode. Rather, the spectra are better understood as reflecting a redistribution of spectral weight toward low energies.

Therefore, the apparent rapid collapse of the fitted phonon energy between 102 K and 97 K does not imply that the mode energy literally drops from ~ 3 meV to zero within a few Kelvin, but rather reflects the fact that the mode becomes strongly damped and merges with the quasi-elastic response as the transition is approached. In this regime we cannot reliably determine a phonon energy, which is why the corresponding points were not included in Figs. 4 (and 5 at the L point).

To avoid any possible misunderstanding, we have softened the wording in the manuscript and clarified that the phonon becomes strongly damped or overdamped close to T_{CDW} , rather than implying a well-defined phonon energy approaching zero.

4. The connection between theory and data should be better explained. For $L=0.5$, two modes are expected in theory, but only one is seen experimentally, which mode does it correspond to in the calculations, and why is the other mode not seen?

Our Answer We thank the reviewer for this important comment. At the L point, the calculations predict two nearby modes of L_2^- symmetry. One is the strongly renormalized branch connected to the harmonic instability and is the mode we identify experimentally as the soft mode. The second nearby L_2^- mode undergoes an avoided crossing with it, so the two branches exchange character and spectral weight as a function of temperature. In the experiment, the two modes cannot be cleanly resolved because their energy separation is comparable to the instrumental resolution and because only one of them carries significant dynamical structure factor in the relevant temperature range, especially at low temperature. The experimentally observed peak therefore corresponds to the L_2^- excitation with the dominant spectral weight, while the second branch remains unresolved.

Dear Editor,

We thank the referee for the careful reading of our revised manuscript and for their positive assessment of the significance of our work. We are pleased that the referee finds the electron-phonon mechanism presented here suitable for publication in *Nature Communications*. We have addressed the remaining minor points as detailed below.

Response to the referee

1. For the error bars, were they determined when all the free parameters were allowed to vary? Sometimes people incorrectly fix “free” parameters when calculating the error bars, which results in very small error bars.

We thank the referee for raising this important point. The error bars were obtained from fits in which all parameters defined as free in the fitting model were allowed to vary simultaneously.

We have clarified this explicitly in the Methods section. In the revised manuscript we now state that the uncertainties shown for the fitted phonon energies correspond to the statistical uncertainties of the full multi-parameter fit, with the elastic intensity, phonon energies, linewidths, and amplitudes varied simultaneously unless explicitly constrained as described in the Methods. The only quantities held fixed were those stated in the manuscript, such as the experimentally measured resolution function and a small number of constrained parameters required to stabilize fits in regions of strong mode overlap. These constrained quantities were not treated as free parameters when estimating the displayed error bars.

2. For the growth of the elastic peak, is this intrinsic, or possibly arises from a distribution of transition temperatures? Even an extremely small fraction of the sample with an elevated T_{CDW} could give a large elastic signal slightly above the mean T_{CDW} . If this is intrinsic, then there is both order-disorder and phonon softening processes, which makes this work consistent with Ref. 45.

We thank the referee for raising this important point. A distribution of transition temperatures would lead to a broadened onset of the CDW elastic intensity. This is not what we observe. On the contrary, thermodynamic, transport, and diffraction measurements on crystals from the same growth show a very sharp CDW transition. The transition width we extracted from the elastic intensity is also extremely small: at the L point, the transition width observed is of the order of 0.2 K, as shown in Fig. S4.

This sharp onset makes it highly unlikely that the elastic intensity above T_{CDW} arises from a small sample fraction with an elevated local transition temperature. For these reasons, we interpret the finite elastic or quasi-elastic signal above the transition as intrinsic, reflecting the development of slowly fluctuating CDW correlations which evolve into true superlattice Bragg peaks below T_{CDW} .

We have revised the discussion to clarify that the transition contains both a displacive soft-phonon component and an order-disorder, or central-peak-like, component associated with slow

CDW correlations. This interpretation is indeed consistent with Ref. 45 and with our observation that the elastic intensity grows most strongly near the L point, where the phonon softening is also largest. The manuscript already shows that the L-point low-energy spectral weight exceeds a simple resolution-limited elastic contribution and is consistent with a softening phonon mode, while the elastic intensity grows most strongly near L on cooling. We have modified accordingly the discussion of Fig. 5c:

“Thermodynamic, transport, and diffraction measurements show a sharp CDW transition (see Supplementary Figure S4) [47], ruling out a sizable distribution of local transition temperatures as the origin of this elastic precursor. Therefore, rather than the ordering of a small fraction of the sample at higher temperature, this enhancement of the elastic line signals the development of intrinsic, slowly fluctuating CDW correlations. As the transition is approached, these fluctuations slow down and grow in spatial extent, giving rise to resolution-limited superlattice Bragg peaks below T_{CDW} . This in turn indicates that, in addition to the displacive, soft-mode character heralded by the pronounced softening of the L point phonon, the CDW transition contains also an order-disorder, central-peak-like component, consistent with Ref. [45].”

3. Soft phonons were also reported in KV_3Sb_5 recently. Are these similar to what is seen here in CsV_3Sb_5 ? If the soft phonons behave differently in the two systems, could these differences be related to the more complex phases and behaviors in CsV_3Sb_5 , such as the $(2 \times 2 \times 4)$ phase, nematicity, and chiral transport?

We thank the referee for pointing out this important recent work. Indeed there are clear similarities between our results on CsV_3Sb_5 and the report of soft phonons in KV_3Sb_5 : pronounced phonon softening is observed near the CDW ordering wave vector, the strongest instability occurs near the L point, and the momentum dependence is more naturally explained by momentum-dependent electron-phonon coupling than by a simple Fermi-surface-nesting scenario. This supports the view that a soft-phonon instability driven by EPC is a common feature of the AV_3Sb_5 family. We have now accordingly included the following text in the discussion:

“Recent IXS measurements on KV_3Sb_5 have also reported soft phonons associated with CDW formation [60]. The similarity to the present CsV_3Sb_5 results suggests that an EPC-driven soft-phonon mechanism is likely common to the AV_3Sb_5 family.”

In addition, we have added an extra panel in Supplementary Figure S7: this reports the temperature dependence of the soft phonon damping ratio extracted from our fits, showing a similar behavior to the one reported for the soft phonon in KV_3Sb_5 .

4. There are many “??” in referencing the SI.

We thank the referee for catching this. We have corrected all unresolved Supplementary Information references and checked that all Supplementary figures, sections, and citations are now properly cross-referenced.